Evaluation of the effect of the application of Quercus cerris extract and the use of fluoride bonding material on the bonding strength of orthodontic brackets after tooth bleaching with hydrogen peroxide

Ay Ezgi
Dursun Derya d_dursun83@hotmail.com
Department of Orthodontics, The University of Health Sciences , Istanbul , Turkey
Serim Ahmet Tansel
Electronic publication date: 2025 Apr 29
Publication date: 2025
Volume: 13
Electronic Location ID: e19335
Received 2024 Nov 12; Accepted 2025 Mar 26
Copyright: ©2025 Ay and Dursun
Copyright year: 2025
Copyright holder: Ay and Dursun
License: This is an open access article distributed under the terms of the Creative Commons Attribution License, which permits unrestricted use, distribution, reproduction and adaptation in any medium and for any purpose provided that it is properly attributed. For attribution, the original author(s), title, publication source (PeerJ) and either DOI or URL of the article must be cited.
License URL: https://creativecommons.org/licenses/by/4.0/

Keywords: Antioxidant, Bleaching, Bond strength, Braces, Fluoride

Funding: Scientific Research Projects Unit of the University of Health Sciences 2021/131 This work was supported by the Scientific Research Project numbered 2021/131 by the Scientific Research Projects Unit of the University of Health Sciences. The funders had no role in study design, data collection and analysis, decision to publish, or preparation of the manuscript.

==============================
The aim of this study was to evaluate the effect of natural antioxidant activity obtained from Q. cerris (Quercus cerris) extract and fluoride-releasing sealant on the shear bond strength (SBS) of orthodontic brackets in teeth after bleaching with hydrogen peroxide (HP). In this research, 200 teeth were divided into two groups, the TT (n = 100) and the OT (n = 100). Each group was further divided into five subgroups (n = 20): TT1 and OT1 = no bleaching; TT2 and OT2 = bleaching; TT3 and OT3 = bleaching+two weeks delayed bonding; TT4 and OT4 = bleaching+10% sodium ascorbate (SA) TT5 and OT5 = bleaching+10% Q. cerris extract. The TT groups were bonded with Transbond sealent + Transbond XT (TT; 3M/Unitek), and the OT groups were bonded with Opal Seal primer (Reliance Orthodontic Products Itasca, IL, USA)+ Transbond XT. Samples were assessed by SBS testing and adhesive remnant index (ARI) scoring. Two-way ANOVA variance analysis, Tukey multiple comparison test and the Chi-square test were used for statistical evaluation. The highest SBS values were obtained in the TT1 (3.17 ± 1.1 MPa) and OT1 (12.58 ± 1.47 MPa) groups, and the lowest SBS values were obtained in the TT2 (4.40 ± 1.11 MPa) and OT2 groups (4.19 ± 1.02 MPa) (p = 0.0001). The mean value of SBS of TT5 was statistically significantly lower than the mean SBS value of TT3 and TT4 (p = 0.042, p = 0.047). No statistically significant difference was observed in the ARI score distribution in the OT groups (p = 0.062), while a significant difference was noticed in the TT groups (p = 0.006). The results indicated that Q. cerris can be safely used to enhance SBS in bleached teeth, based on the dose, duration, and application procedure parameters utilized in this study. In addition, the fluoride-releasing sealant provides clinically sufficient SBS.

Introduction

Tooth bleaching is a fast, conservative, and economical aesthetic treatment option preferred by dentists and individuals for the treatment of discolored teeth (Suneetha et al., 2014). However, free radicals and residual oxygen absorbed in the enamel prisms and dentin structure immediately after the completion of bleaching treatment inhibit resin polymerization and cause a decrease in the shear bond strength (SBS) of orthodontic brackets (Ambersari, Karunia & Alhasyimi, 2024; Vidhya et al., 2011). Recommendations to delay restorations have ranged from 24 h to 3 weeks to eliminate the reduction in SBS after bleaching. However, there is no agreement on the delay time in the literature (Miljkovic et al., 2024; Arumugam et al., 2014; Souza-Gabriel et al., 2011; Sundfeld et al., 2005). In addition, an in vitro study has shown that bleaching treatment performed before starting orthodontic treatment is more effective and gives more significant results than bleaching after debonding (Hintz, 2001).

Several methods have been proposed to reverse the decreased SBS caused by bleaching, including treating the bleached enamel surface with alcohol, using adhesives containing organic solvents, removing the outermost layer of enamel, and applying antioxidants (Hobby et al., 2012; Vidhya et al., 2011). Sodium ascorbate (SA) is recognized as a biocompatible, neutral, and effective antioxidant, evidenced by its protective role against free radical damage and its contribution to improved shear bond strength (Taib et al., 2020). However, it has also been demonstrated that SA, as a synthetic antioxidant, may possess mutagenic potential in mammalian somatic cells (Chen et al., 2022). Therefore, interest in natural antioxidants of plant origin, such as pine bark extract, green tea, pomegranate peel extract, aloe vera, and grape seed extract, has been increasing in recent years (Güler et al., 2013; Najib et al., 2021).

Quercus cerris is a variety of the hairy oak species. Long used in traditional medicine, Quercus species have been reported to have a broad spectrum of biological effects such as antioxidant, antidiabetic, anticancer, antiproliferative, antimicrobial, gastroprotective, anti-inflammatory, and antibacterial. It has therefore been used for numerous purposes in medicine, including the treatment of bleeding, dysentery, chronic diarrhea, gastritis, asthma, pyrexia, Parkinson’s disease, and gum problems (Hobby et al., 2012; Najib et al., 2021; Pinto et al., 2019; Taib et al., 2020). Despite the medicinal value of Quercus species, very few studies of it have been found in the dental field.

Fluoride-releasing bonding materials are commonly used in orthodontics to prevent white spot lesions. There is a lack of sufficient studies in the literature that investigate the effects of these materials on bracket bond strength, especially in relation to bleached teeth. Teeth whitening procedures, while effective for aesthetic enhancement, can negatively impact the bonding strength of orthodontic brackets due to residual free radicals and oxygen within the dental structure (Cengiz-Yanardag & Karakaya, 2024; Vidhya et al., 2011). Although synthetic antioxidants like sodium ascorbate have shown promise in mitigating this effect, their potential health risks and lower efficacy compared to natural antioxidants limit their widespread clinical application (Barteková et al., 2021; Taghvaei & Jafari, 2015).

Despite emerging interest in natural antioxidants, no clinically viable protocol has been developed for their use in whitening treated teeth to restore bond strength effectively. Additionally, while fluoride-releasing bonding agents have demonstrated remineralization benefits, their potential impact on bracket bond strength remains underexplored, particularly in the context of bleached enamel. Addressing these gaps, this study aimed to evaluate the effects of different natural antioxidant treatment procedures, emphasizing the antioxidant properties of acorn extract and fluoride-releasing bonding materials on the SBS of orthodontic brackets on bleached teeth. The research question is: What are the effects of different natural antioxidant treatments and fluoride-releasing bonding materials on the SBS of orthodontic brackets on bleached teeth, and how do these effects differ when compared to conventional bonding materials?

Materials and Methods

Preparation of samples

This study was approved by the Scientific Research Committee of the University of Health Sciences, Istanbul, Turkey (25.06.2021/22/11). According to the statistical power analysis (G*Power ver. 3.1.9.4; Franz Faul, Universität Kiel, Germany) the effect size of the groups was determined as 0.40 and the sample size analysis performed with a power value of 0.80 resulted in a sample size of 20 for each group at a significance level of p = 0.05.

There are some disadvantages and limitations associated with the use of human teeth for in vitro studies. Human teeth may have extensive caries lesions, and repair to caries stimuli may impair standardization in determining the bond strength of dentin. It may also be difficult to check the origin and age of the collected teeth. The relatively small and curved surface area of human teeth may also be a limitation for specific tests that require flat surfaces of uniform thickness (Mellberg, 1992; Zero, 1995). Therefore, bovine teeth were used in our study.

Two hundred bovine incisors from heads were obtained from an abattoir where cattle are slaughtered. The criteria for selection were the absence of fractures, caries, and fissures on the buccal enamel surface and the absence of any malformation on the surface of the teeth (Mellberg, 1992; Zero, 1995).

Teeth were randomly divided into 10 groups according to treatment protocols and 20 teeth were included in each subgroup: TT1 and OT1 = no bleaching; TT2 and OT2 = bleaching; TT3 and OT3 = bleaching followed by a 2-week delay before bonding; TT4 and OT4 = bleaching with the application of 10% sodium ascorbate (SA); and TT5 and OT5 = bleaching with the application of 10% Q. cerris extract. The first five subgroups (TT) (n = 100) were bonded using Transbond sealant and Transbond XT (TT; 3M/Unitek), while the second five subgroups (OT) (n = 100) were bonded using Opal Seal primer (Reliance Orthodontic Products Itasca, IL, USA) and Transbond XT. The specimen distribution along with the materials used is summarized in Table 1.

The samples were first cleaned with a periodontal curette under running water to remove debris. They were then brushed with a micromotor using a soft hairbrush and fluoride-free fine pumice at low speed for 15 s, washed for 15 s, and then dried. The samples were disinfected in 0.1% thymol solution for one week. They were kept in distilled water until the time of the experiment (Ajlouni, Bishara & Oonsombat, 2004).

Bleaching procedure

Office bleaching agent Opalescence Boost PF (Ultradent, Inc., South Jordan, UT, USA) containing 40% hydrogen peroxide (HP) was applied to all samples according to the manufacturer’s recommendations except the ones in groups TT1 and OT1. Approximately one mm thick bleaching gel was applied on the buccal surface of the teeth for 20 min. Then, the gel was gently removed from the tooth surface with a sponge and the surface was washed with distilled water for 60 s. The bleaching procedure was applied to each tooth surface a total of three times (Aksakalli, Ileri & Karacam, 2013).

Table 1 The distribution of specimens and study groups (n = 20).

TT (n = 20)/OT (n = 20) groups	Bleaching treatment	Antioxidant treatment	Bracket attachment	
Groups TT1 and OT1	–	–	Done immediately	
Groups TT2 and OT2	% 40 Hydrogen peroxide 20 min × 3 times	–	Done immediately	
Groups TT3 and OT3	% 40 Hydrogen peroxide 20 min × 3 times	–	Done 2 weeks after bleaching	
Groups TT4 and OT4	% 40 Hydrogen peroxide 20 min × 3 times	%10 Sodium Ascorbate for 10 min	Done immediately	
Groups TT5 and OT5	% 40 Hydrogen peroxide 20 min × 3 times	%10 Q. cerris extract, for 10 min	Done immediately	

Preparation of antioxidant solutions

Preparation of sodium ascorbate antioxidant

A 10% SA solution was prepared by dissolving 3 g of powdered SA (Sigma-Aldrich, St. Louis, MO, USA) in 30 ml of distilled water and stored in a tightly closed glass jar and refrigerator at 4 °C until the time of the experiment and used within a maximum of 1 week (Güler et al., 2013).

Q. cerris sample collection and preparation of antioxidant

Quercus cerris is a Mediterranean oak tree that thrives under Mediterranean climatic conditions, and its flowering time is September–October in the autumn (Najib et al., 2021). For this study, wild Q. cerris fruit was harvested in October 2021 from a forest that is free from traffic and heavy metals. The botanical identification of the leaves, bark, and acorns was carried out using international standard analyses. All samples were deposited in the herbarium of Hacettepe University (HUB) under the voucher specimen number “E. Çilden 1976 (Quercus cerris L.)”.

Crude extraction of phenolic compounds was carried out using maceration (Chen et al., 2022; Gülçin, 2005). The Q. cerris fruits were then powdered with the help of a blender, and forty grams of powdered fruit was added to 200 mL of 96% ethanol solution and extracted at 37 °C for 24 h. The extracted solution was collected and the residues were extracted three times under the same conditions. The combined solution was concentrated at 60 rpm at 40 °C using a rotary evaporator (model RV 8; IKA, Shanghai, China) to obtain the ethanol crude extract of Q. cerris. The obtained crude extract weighed 3.93 g, and 39.3 mL of 96% ethanol was added, placed in an ultrasonic bath, and kept at ultrasonic vibration for 15 min and stored in amber bottles at 4 °C (Chen et al., 2022; Gülçin, 2005).

Application of antioxidants

The TT4 and OT4 groups were treated with 10% SA antioxidant solution, and the TT5 and OT5 groups were treated with 10% natural antioxidant solution obtained from Q. cerris extract. Antioxidant solutions were applied to enamel surfaces for 10 min each and the solution was refreshed at the end of each minute. Then, the samples were washed with distilled water for 20 s and the brackets were bonded.

Bracket bonding procedure

In our study, 200 upper central brackets (Mini Twin; RMO, Denver, CO, USA) with a 0.018 inch slot, 80-gauge mesh foil base, and 14.2 mm2 surface area were used. All the enamel surfaces of the teeth were dried and 37% phosphoric acid (Panora 200; Imicryl, Konya, Turkey) was applied for 30 s. Then the surfaces were washed and dried with an air syringe for 15 s. A thin layer of Transbond sealant (3M/Unitek, Monrovia, CA, USA) was applied to the samples of the TT groups and irradiated for 5 s (O-Led, Guilin Woodpecker, Guangxi, China). A thin layer of Opal Seal adhesive primer (Opal Orthodontics, Ultradent, South Jordan, UT, USA) was applied to the samples of OT groups and irradiated for 5 s. Transbond XT adhesive paste was placed on the base of the brackets. All the brackets were placed at the center of the crowns and the adhesives were cured mesially and distally for 10 s each (see Fig. 1).

Figure 1 Bonded brackets and embedded bovine incisors into acryl resin.

Thermal cycle

All samples were thermocycled at 1,000 cycles with 30 s dwell time and 5 s transfer time in distilled water baths at 5 ± 2 °C and 55 ± 2 °C, respectively (Thermocycler, model THE-1100, SD Mechatronik, Feldkirchen-Westerham, DE) (Fletcher-Stark et al., 2011).

SBS analysis

The samples were embedded in autopolymerizing cold acrylic resin placed in molds with the long axis perpendicular to the ground, one mm apical to the enamel–cementum junction for ease of numbering and placement in the test device (see Fig. 1).

SBS tests of the brackets were evaluated with a universal testing machine (model Autograph AGS-X; Shimadzu, Kyoto, Japan). Acrylic blocks were fixed to the test device. Before the test, the parallelism of the base of each bracket and the edge of the device was checked. Force was applied to the tooth-bracket interface at a rate of one mm/min until the bracket debonded. The force at the moment of bracket debonding was recorded as measured in Newtons (N) using a computer connected to the device. The results were converted to megapascals (MPa) by dividing the debonding force by the bracket base area.

Adhesive remnant index

The enamel surfaces of the teeth were evaluated with a stereomicroscope (model SZX10; Olympus, Tokyo, Japan) with a magnification of 20X. The adhesive remaining on the tooth surface was graded using the adhesive remnant index (ARI) developed by Årtun & Bergland (1984). According to this classification, Score 0 means no adhesive remains on the tooth surface, Score 1 means less than 50% of the adhesive is remaining on the tooth surface, Score 2 indicates that more than 50% of the adhesive is remaining on the tooth surface, and Score 3 shows that all of the adhesive remains on the tooth surface (see Fig. 2).

Figure 2 The stereomicroscope images of the ARI scores of the teeth (A: Score 0, B: Score 1, C: Score 2, D: Score 3).

Statistical analysis

SPSS for Windows, version 15.0 (SPSS Inc., Chicago, IL, USA) was used for all statistical analyses. The analysis of variance (ANOVA) test was used to analyze the mean differences among the groups. The results of the ANOVA demonstrated a difference between the groups; therefore, the post hoc Tukey test was performed to explore the significance. The Chi-square test was used to compare the qualitative data of ARI scoring. Significance level was determined as p < 0.05.

Results

The results of the ANOVA of the SBS values of the TT and OT groups that were bonded using two different sealants are shown in Table 2, and the results of the Tukey multiple comparison test are shown in Table 3.

Table 2 Comparison of mean SBS (Mpa) values between experimental groups.

	TT
(mean ± SD)	OT
(mean ± SD)	p	
Groups TT1-OT1	13.17 ± 1.10	12.58 ± 1.47	0.156	
Groups TT2-OT2	4.40 ± 1.11	4.19 ± 1.02	0.542	
Groups TT3-OT3	11.36 ± 2.03	10.53 ± 2.01	0.201	
Groups TT4-OT4	11.33 ± 2.30	10.46 ± 2.08	0.213	
Groups TT5-OT5	9.51 ± 3.06	9.07 ± 2.24	0.607	
Notes.

Statistically significant (p < 0.05). SD: Standard deviation.

Table 3 P values of pairwise comparison of SBS values of five groups between groups.

Tukey multiple comparison test	TT /	OT	
Group TT1-TT2/Group OT1-OT2	0.0001*	0.0001*	
Group TT1-TT3/Group OT1-OT3	0.046*	0.005*	
Group TT1-TT4/Group OT1-OT4	0.045*	0.003*	
Group TT1-TT5/Group OT1-OT5	0.0001*	0.0001*	
Group TT2-TT3/Group OT2-OT3	0.0001*	0.0001*	
Group TT2-TT4/Group OT2-OT4	0.0001*	0.0001*	
Group TT2-TT5/Group OT2-OT5	0.0001*	0.0001*	
Group TT3-TT4/Group OT3-OT4	0.999	0.998	
Group TT3-TT5/Group OT3-OT5	0.042*	0.09	
Group TT4-TT5/Group OT4-OT5	0.047*	0.121	
Notes.

* Statistically significant (p < 0.05).

These descriptive statistics show the variation in SBS, with the highest bond strength in the TT1 (13.17 ± 1.10 MPa) and OT1 (12.58 ± 1.47 MPa) groups and the lowest bond strength in the TT2 (4.40 ± 1.11 MPa) and OT2 (4.19 ± 1.02 MPa) groups. No statistically significant differences were observed in terms of SBS between the TT and OT groups (see Table 2). The Tukey test showed that there were significant differences between the SBS of all TT groups except the groups TT3 and TT4, OT3 and OT4, OT3 and OT5, OT4 and OT5 (see Table 3). ARI score distributions of the TT and OT groups were compared, and the results are shown in Tables 4 and 5.

Table 4 P values of pairwise comparison of adhesive remnant index (ARI) scores of five groups between groups.

	TT	
Group TT1-TT2	0.008*	
Group TT1-TT3	0.944	
Group TT1-TT4	0.626	
Group TT1-TT5	0.254	
Group TT2-TT3	0.002*	
Group TT2-TT4	0.01*	
Group TT2-TT5	0.003*	
Group TT3-TT4	0.506	
Group TT3-TT5	0.150	
Group TT4-TT5	0.053	
Notes.

* Statistically significant (p < 0.05).

Table 5 Comparison of adhesive remnant index (ARI) scores.

		Groups
TT1 and OT1	Groups
TT2 and OT2	Groups
TT3 and OT3	Groups
TT4 and OT4	Gr o ups
TT5 and OT5	p value	
TT	Score 0	2	10.00%	10	50.00%	1	5.00%	2	10.00%	5	25.00%	0.006*	
Score 1	7	35.00%	8	40.00%	7	35.00%	10	50.00%	3	15.00%	
Score 2	10	50.00%	2	10.00%	11	55.00%	8	40.00%	12	60.00%	
Score 3	1	5.00%	0	0.00%	1	5.00%	0	0.00%	0	0.00%	
OT	Score 0	4	20.00%	12	60.00%	4	20.00%	3	15.00%	7	35.00%	0.062	
Score 1	7	35.00%	7	35.00%	6	30.00%	8	40.00%	6	30.00%	
Score 2	9	45.00%	1	5.00%	9	45.00%	9	45.00%	7	35.00%	
Score 3	0	0.00%	0	0.00%	1	5.00%	0	0.00%	0	0.00%	
	P value	0.633	0.748	0.557	0.786	0.266		
Notes.

* Statistically significant (p < 0.05).

The ARI score distributions of the TT and OT groups were compared and evaluated. No statistically significant differences were observed between the ARI score distributions of the TT and OT groups. However, significant differences were observed between the ARI score distributions of the TT groups (p < 0.05) (see Table 4). The ARI Score 2 and Score 3 distributions of group TT2 were significantly lower than the ARI Score 2 and Score 3 distributions of group TT1, TT3, TT4, and TT5 (p < 0.05). No statistically significant differences were observed between the ARI score distributions of the OT groups (see Table 5).

Discussion

For orthodontic treatment to be successful, the attachments must be bonded with a retention that can withstand the forces occurring in the mouth. During orthodontic treatment, there is a need for solutions that will provide the desired level of remineralization, are practical, are easy to apply, and offer advantages such as long-term fluoride release, and will not adversely affect the bracket strength. The number of studies examining the effects of fluoride-releasing bonding agents on bracket bond strength is insufficient in the literature, nor have the effects of fluoride-releasing bonding agents on bleached teeth been extensively investigated. In addition, the application of antioxidant agents is a reliable method that strengthens bonding by neutralizing free oxygen radicals and eliminates the waiting period. Although there is a wide variety of natural plant extracts, so far no antioxidant treatment protocol has been developed in a form suitable for clinical use.

As a result of this study, SBS values were found to be lower in the bleaching group due to the strong oxidative effect of hydrogen peroxide (TT2 and OT2 groups). These results prove that bleaching treatment has a negative effect on bonding, and our results are thus similar to several other studies (Alatta et al., 2024; Brock et al., 2024; Whang & Shin, 2015). The reason for the decrease in bond strength after bleaching is that the residual oxygen and free radicals released as a result of the decomposition of HP prevent the infiltration and polymerization of the resin (Bittencourt et al., 2010; Sung et al., 1999). In addition, some researchers have suggested that changes in the enamel structure, loss of prismatic shape, increased porosity, and dentin permeability that occur after bleaching are effective in the weakening of the resin–enamel bond strength (Swapna et al., 2014; Türkün & Kaya, 2004).

It has been reported that delaying bracket bonding for periods between 24 h and 3 weeks after bleaching treatment removes the structural changes caused by free oxygen radicals in the enamel tissue (Bittencourt et al., 2010; Lai et al., 2002; Arumugam et al., 2014). In our study, the mean values of SBS were statistically significantly higher in the group in which the bracketing phase was performed after waiting two weeks following the bleaching treatment (TT3 and OT3) than in the group in which the bracketing phase was performed immediately after the bleaching treatment (TT2 and OT2) (see Table 3). These findings are similar to of those other studies reporting an increase in the resin–enamel bond strength at the end of a two-week clinical waiting period (Mathai et al., 2015; Whang & Shin, 2015).

In studies in which relatively low concentrations of hydrogen peroxide were used as a bleaching agent or bracket placement was delayed for a longer period of time, SBS values in the delayed group were close to the control group (Bittencourt et al., 2010; Mazaheri et al., 2011; Whang & Shin, 2015). However, in our study, the SBS values in the groups that were kept for two weeks after bleaching before bracketing (TT3 and OT3) were lower than those in the control groups without bleaching (TT1 and OT1). It is thought that the reason for this may be related to the high concentration of HP used in the study. In accordance with our study findings, Bittencourt et al. (2010) stated that the SBS of the group in which a 21-day delay period was applied and the control group were statistically similar in enamel and dentin samples bleached with 35% HP, but the SBS of the groups in which 7- and 14-day delay periods were applied could not reach the average SBS value of the control group.

Various methods have been used after bleaching treatment to increase bond strength without long waiting periods. Recently, there have been many studies aiming to increase the bonding value by applying antioxidants to the tooth surface after bleaching, which neutralizes free oxygen radicals and eliminates the negative biological effects of peroxides (Brock et al., 2024; Grazioli et al., 2024; Pauletto et al., 2024). To date, antioxidant applications to the tooth surface after bleaching treatment have been tested to increase bond strength (Arumugam et al., 2014; Da Silva et al., 2011; Whang & Shin, 2015). A number of in vitro studies presented an increased bond strength with sodium ascorbate antioxidant treatment. Researchers reported that 10% SA can be an alternative to the two-week clinical delay (Danesh-Sani & Esmaili, 2011; Güler et al., 2013; Kimyai & Valizadeh, 2006; Lai et al., 2002; Türkün & Kaya, 2004). Our results are consistent with these studies; while the SBS values in the SA and control groups followed a similar trend, Table 3 indicates a statistically significant difference between TT1 and TT4. However, it should be kept in mind that SA antioxidant treatment protocols applied to bleached enamel are not standardized in these studies, and differences exist between them in concentration, form, application times, and methods (Freire et al., 2011; Lai et al., 2002; Nascimento et al., 2016; Park, Kwon & Kim, 2013).

In our study, 10% SA solution was actively applied to the bleached enamel surfaces for 10 min, and the solution was refreshed at the end of each minute (see Table 1). The SA antioxidant treatment we applied to the bleached enamel surfaces increased the bond strength, and sufficient SBS values were obtained for clinical use, similar to several in vitro studies (Danesh-Sani & Esmaili, 2011; Güler et al., 2013; Kimyai & Valizadeh, 2006; Lai et al., 2002; Türkün & Kaya, 2004). However, it was observed that the SA treatment did not restore SBS values to the level of the control group. The microanatomy and prism dimensions of the bovine teeth that were used in this study differ from human dentition (Olek, Klimek & Bołtacz-Rzepkowska, 2020). Due to this difference, the penetration of SA antioxidant into the enamel might occur more slowly. Increasing the duration of antioxidant application may be an alternative that could achieve the SBS of the control group.

Although natural plant extracts and antioxidant solutions obtained from them show diversity in studies, no natural antioxidant treatment protocol has been developed in a form suitable for clinical use and has taken its place in routine dental practice (Feiz, Mosleh & Nazeri, 2017; Garcia et al., 2012; Rana et al., 2019; Satue, Huang & Frankel, 1995). Organic extracts of Q. cerris have strong antioxidant and antimicrobial properties due to their richness in bioactive phenolic compounds such as tannins and flavonoids (Vinha et al., 2016).

The SBS value measured after treatment of bleached enamel surfaces of the Q. cerris groups was higher than that of the groups that had only bleaching (see Table 2). These findings suggest that residual oxygen can be neutralized and the reduced SBS can be reversed by the application of the experimental antioxidant obtained from Q. cerris extract (Årtun & Bergland, 1984).

In the study, the mean SBS values of the bleached enamel surfaces after experimental antioxidant treatment obtained from Q. cerris extract were found to be significantly lower compared to the groups treated with SA-containing experimental antioxidant treatment (see Table 3). This may be related to the fact that the antioxidant activity of SA was stronger than the experimental antioxidant obtained from Q. cerris extract. SA, as a compound with high antioxidant capacity, can effectively neutralize free radicals formed during bleaching. This neutralization contributes to the reduction of oxidative stress on the enamel surface after bleaching and the removal of residual oxygen that negatively affects bond strength (Ghaleb et al., 2020; Moosavi et al., 2024; Pauletto et al., 2024). In contrast, Q. cerris extract contains natural antioxidants such as phenolic compounds and tannins. Although the antioxidant properties of these compounds are potent, they may show a more limited activity compared to sodium ascorbate in terms of enamel surface penetration or oxygen neutralization. To evaluate the potential of Q. cerris extract in clinical applications, additional studies investigating the efficacy of different concentrations, application times, and combination treatments are needed. In particular, optimizing the potential of this natural antioxidant to increase the bond strength to the bleached enamel surface may encourage its clinical use.

In this study, Opal Seal was used as the fluoride-releasing agent. It can be concluded that the effect of fluoride-releasing varnishes on bleached teeth remains unchanged for different bonding strength-enhancing methods, rendering both statistical and clinical these methods ineffective. Furthermore, another possible reason for the ineffectiveness of fluoride-releasing materials is the limited role of these materials on bonding strength. Fluoride-releasing materials are mainly designed for caries prevention and remineralization of the enamel surface. However, fluoride release may not be sufficiently effective in neutralizing oxygen radicals in the microstructure of the enamel surface. This may adversely affect resin penetration after bleaching and compromise the effectiveness of other methods to improve bond strength (Bittencourt et al., 2010). In a study conducted by Yetkin & Sayar (2020), the effect of different bonding materials on SBS was examined and no statistically significant difference was found between the Opel Seal group and the other groups. This result is in parallel with the findings in our study.

Accurate ARI scoring is an important factor in the choice of orthodontic adhesive. ARI scores determine where and how mechanical retention occurs during the bracket debonding process, depending on whether the adhesive remains on the tooth surface, at the bracket base or within the adhesive itself. In ARI scores, a score of 3 indicates adequate mechanical retention between the adhesive and the tooth, whereas a score of 0 indicates adequate mechanical retention between the bracket and the adhesive, and other scores indicate cohesive rupture (rupture within the adhesive itself) (Faltermeier et al., 2007).In this study, the low presence of ARI Score 2 and Score 3 in the TT2 group compared to the other groups may be related to the SBS values of this group. Low SBS values indicate that the bonding adequacy between the adhesive and the tooth surface is weakened, and rupture occurs mostly at the adhesive–tooth interface. Although the bonding materials used in the study were different, the use of the same brackets and the main bonding material (e.g., Transbond XT) may have been effective in the similarity of the ARI scores of the other groups. The base design of the brackets plays an important role in bond strength and rupture mechanism. The use of the same bracket design may have resulted in similar mechanisms of rupture at the bracket–adhesive interface. However, the fact that the ARI scores were generally centered on Score 1 and Score 2 suggests that the rupture occurred mostly at the adhesive–tooth interface or within the adhesive and that the bond strength met a certain threshold.

The limitations of this study are that the experiments were performed in vitro, only certain concentrations of antioxidants were used, and only a single bleaching agent and fluoride-releasing binder were used. In addition, the effects of antioxidant treatments on different application times were not evaluated and the potential side effects and biocompatibility of Q. cerris extract were not investigated. These factors suggest that caution should be exercised in generalizing the findings to clinical applications. Future studies may provide more comprehensive data by addressing these limitations.

Conclusion

Bleaching treatment significantly reduces the SBS of brackets, but antioxidant treatments increase the SBS of brackets. While Q. cerris extract may not increase SBS to the same extent as SA, it remains a favorable choice for application in bleached teeth due to its natural antioxidant feature. However, further clinical and laboratory studies are necessary to assess the impact of various concentrations of Q. cerris extract and to identify potential side effects before its application in routine practice can be considered. In addition, since fluoride-releasing sealants show similar SBS values to the conventional system, they can be preferred in daily applications in bleached teeth.

Supplemental Information

Supplemental Information 1 Pictures of experimental process

Supplemental Information 2 Raw data

The authors thank Associate Professor Emre Çi̇lden, Serhat Ozmen, Assistant Professor Emir Alper Türkoğlu and Assistant Professor Tuğba Hali̇loğlu Özkan for their help in this research.

Additional Information and Declarations

Competing Interests

Author Contributions

Human Ethics

Data Availability

The authors declare there are no competing interests.

Ezgi Ay conceived and designed the experiments, performed the experiments, analyzed the data, prepared figures and/or tables, authored or reviewed drafts of the article, and approved the final draft.

Derya Dursun conceived and designed the experiments, analyzed the data, prepared figures and/or tables, authored or reviewed drafts of the article, and approved the final draft.

The following information was supplied relating to ethical approvals (i.e., approving body and any reference numbers):

This study was approved by the Scientific Research Committee of the University of Health Sciences, Istanbul, Turkey (25.06.2021/22/11).

The following information was supplied regarding data availability:

The raw data and experimental pictures are available in the Supplemental Files.

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
