# Peer review of "Evaluation of the effect of the application of Quercus cerris extract and the use of fluoride bonding material on the bonding strength of orthodontic brackets after tooth bleaching with hydrogen peroxide"

_PeerJ, doi:10.7717/peerj.19335_

## Round 0.1 · original submission · Major Revisions

Medicinal plants hold significant promise as natural sources of antioxidants. I believe your research has the potential to make a valuable contribution to your colleagues. To further enhance the clarity and impact of your article, I recommend addressing certain technical aspects. I strongly encourage you to carefully review all reviewer suggestions and give each one thoughtful consideration. If you disagree with any particular suggestion, it would be beneficial to provide clear and well-reasoned justifications for your perspective.

Reviewer 1 ·

Basic reporting

• The complete manuscript requires revision by a native English speaker. Numerous grammatical problems require correction. Kindly submit the certificate of proofreading by the professional along with the revisions.
• The title and the background are not well-related. The title outlines the evaluation of Quercus cerris extract's efficacy of improving the bond strength of composite following dental bleaching with hydrogen peroxide. The title does not reflect the study's focus on the shear bond strength (SBS) of orthodontic brackets, despite the aim of the study and the entire paper discussing the evaluation of the effects of natural antioxidant activity from Quercus cerris extract and fluoride-releasing sealant on the SBS of orthodontic brackets in teeth post-bleaching with hydrogen peroxide.
• Your introduction needs additional elaboration. I recommend enhancing the description, as we did not identify any research gap to substantiate your study; specifically, you should elaborate on the knowledge gap being addressed. You should add that bleaching is more effective and provides more significant results when performed before using orthodontic brackets rather than after debonding. Unfortunately, bleaching may lead to a decrease in the bond strength of orthodontic brackets.
• The figure is relevant but of low quality; please enhance its quality. Additionally, include a figure illustrating the adhesive remnant index under a stereomicroscope with a clear presentation
• In a previous study, it was discovered that using natural extract as an antioxidant after bleaching successfully reverses the reduced shear bond strength of orthodontic brackets after bleaching, so what is new in this study? the use of fluoride-releasing bonding materials? Please evaluate this part of your background.
• Some references are outdated; please utilize the updated literature. We believe that SBS studies, particularly in orthodontics, have significantly advanced in recent years

Experimental design

• What about utilizing bovine incisors instead of human premolars? You need to mention in the method
• The research question is inadequately delineated. Kindly elucidate the research gap comprehensively.
• It is written: The adhesive residual on the tooth surface was evaluated utilizing the Adhesive Residue Index (ARI) established by Årtun and Bergland (1984). Kindly provide a comprehensive description.
• It is inscribed: The Chi-square test for qualitative data comparison was employed at a significance level of p<0.05. What type of qualitative data? ARI?

Validity of the findings

• The discussion chapter holds the utmost significance in a paper; nonetheless, the composition of the discussion in this study needs substantial enhancement. The author reiterates the data in the discussion and fails to elucidate how Q. cerris enhances the SBS of orthodontic brackets post-bleaching. Moreover, the discourse of ARI scores and procedures among the five examined groups remains ambiguous. The quality of the debate writing is subpar.

Reviewer 2 ·

Basic reporting

The literature review is sufficient and provides the information to understand the context of the study. The article's structure is appropriate with the provided figures. While the English is understandable, there are a few mistakes and typos to be corrected. The hypothesis for the research is not posed correctly (use of “may”) and is not quite clear. Further, the authors need to revisit their hypothesis in their conclusions, stating whether the hypothesis was correct of incorrect based on the results.
Minor orthographic and grammatical issues that need to be addressed:
1) Please make sure to leave a space between a word and the bracket from a citation
2) Line 71: "Also, to evaluate...": sentence needs to be improved and better connected to the previous sentence.
3) Line 73: what you suggest is not a null hypothesis. You would have such if your hypothesis were that there is no effect by using the agent you propose.
Line 104: Is "sn" supposed to stand for seconds [s]?
Line 119: the unit for temperature is "°C". Please revise.
Line 140: Previously, the notation of T4/O4 was used, here it switched to T4-O4. Please stay consistent.
Line 170: either "until bracket debonding" or "until the bracket debonded"
Line 206: please correct the p value
Line 216: “HP” has not been defined beforehand, but as this is used only twice across the manuscript, I suggest to not abbreviate it at all
Line 224: “to of those other” should read “to those of other”?
Line 269: “but antioxidant treatments increase the reduction in SBS”, does that mean an additional treatment with antioxidants further reduces the SBS? This part is not easy to understand, and I suggest rewriting with more clarity what the antioxidants cause.
Line 275: should write “sealants”

Table 2: Does column 4 show the p values, and you p-value for statical significance is p<0.05, i.e., there is no statistically significant difference in any of the mean values? Please clarify by updating the table and its caption.
Table 3, 4, and 5 (Lines 14, 21, and 28, respectively): apparently you intended to mark statistically significant values with an asterisk (*). Please do so.
What do the values in Table 3 represent? Is it MPa (as the caption suggests)? i.e., is the difference between T1/O1 and T2/O2 in the TT subgroup 0.0001 MPa, as for many other comparisons?
The same issue in Table 4: The caption states the comparison of ARI values. Apparently, the values may be p values from the comparison. The caption needs to clarify what has been done here. Table 5 has some additional information in the table, so it is somewhat clearer, but still there is room for improvement.

In general, the table captions should explain the tables. Please update them.

Experimental design

As mentioned in 1. Basic reporting, the research question and hypothesis need some better definition.
Further, the naming of the groups is confusing with T1/O2 and additional TT and OT - I assume I could read T1/O1 actually as TT1 and OT1, or just as "Group 1", as the subgroups TT and OT are listed with their respective values.
Aside from the confusing naming, the research approach and selected treatments between groups and subgroups within each group seem appropriate.

Validity of the findings

The conclusions made from the results are appropriate, however, they need to be put in context with the original research question/hypothesis. The preference for Q. cerris over sodium ascorbate despite the lower efficacy is not quite understandable and should be better supported. Further, the introduction mentions that synthetic antioxidants have lower efficacy than natural ones, so why do we see the contrary here?

---

## Round 0.2 · Minor Revisions

I appreciate your constructive attitude toward the reviewers' suggestions and improving your article based on their suggestions. Although your article has been revised according to their suggestions, it needs some improvements. Carefully consider each recommendation, assessing its relevance and potential to enhance your work. If you disagree with any specific suggestion, it is important to provide clear and well-reasoned justifications, supported by evidence where applicable, to prove your perspective.

Reviewer 2 ·

Basic reporting

The quality of the English used in the manuscript has much improved. References have been updated with newer studies. However, there are some issues that still need to be addressed.

In lines 203-205, the authors claim that "there were significant differences between the SBS of all TT groups except the groups TT3 and TT4 and OT3 and OT4 (see Table 3)." Table 3 shows no significant differences also for comparisons OT3-OT5 and OT4-OT5. Why are these excluded here?

Within the discussion the term “fluorine” has been used several times instead of “fluoride” as if these terms were interchangeable. While related, they are not the same. Opal Seal is listed as a fluoride-releasing agent. Please correct.

in line 276: "However, it was observed that the control group did not achieve the bonding values." Which bonding values were not achieved? Those listed in other publications? There does not seem to be a value mentioned that was expected to be achieved. Within your own study, the control group achieved the highest SBS values, according to Table 2. What also could have been meant is that after the SA treatment, SBS values as high as those of the control group were not achieved, meaning that although potent, SA did not restore full SBS values. On the other hand, in line 237 the authors state that the SBS from SA treated teeth and control group "were similar". This is ambiguous, as it can be read as there is basically no difference. As table 3 shows a stat. sign. difference between TT1 and TT4, the statement in line 237 needs to be made clearer. In line 276, please clarify which bonding values were not achieved.

Experimental design

previous issues haven been addressed.

Validity of the findings

no comment

---

## Round 0.3 · accepted · Accept

I would like to thank you for accepting the referees' suggestions and improving your article based on their suggestions. Your article is ready to publish. We look forward to your next article.

Reviewer 2 ·

Basic reporting

all issues addressed

Experimental design

no comment

Validity of the findings

no comment

Additional comments

no comment